# Friends or Frenemies? The Role of Social Technology in the Lives of Older People

**DOI:** 10.3390/ijerph16244969

**Published:** 2019-12-06

**Authors:** Tina ten Bruggencate, Katrien G. Luijkx, Janienke Sturm

**Affiliations:** 1Department of Tranzo, School of Social and Behavioral Sciences, Tilburg University, 5037 AB Tilburg, The Netherlands; k.g.luijkx@uvt.nl; 2Chair of People and Technology, Institute for HRM and Psychology, Fontys University of Applied Science, 5612 AR Eindhoven, The Netherlands; j.sturm@fontys.nl

**Keywords:** social technology, social life, social needs, older people, interventions

## Abstract

By having a healthy and happy social life, social needs are fulfilled. When social needs are not fulfilled, loneliness and social isolation can occur, which have negative consequences for one’s physical and mental health. Social technology, technology that enables social interaction, can be a resource to fulfil the social needs of older people. In this study, we aimed to learn what role social technology plays in the social life of older people. We held 15 interviews with people aged over 70 who regularly use some form of social technology. Our results indicate that social technology plays different roles in the lives of older people. It strengthens the existing social relationships and social structures. It also brings depth and fun to the social contacts of older people and in this way, enriches their social lives. Social technology also gives a sense of safety and peace of mind to the older people themselves but also to their network members. However, there are barriers in the use of social technology. The older people struggled with using social technology and feel that social technology sometimes stands in the way of real human contact. In supporting and facilitating people’s relationship with others, a community and society, technology helps fulfil older people’s need for connectedness, meaningfulness and independence. However, the relationship with independence is ambiguous. Their life experience gives older people a thoughtful way of looking at social technology and the role it plays in their lives.

## 1. Introduction

Older people see their social life and their social relationships as important prerequisites for successful ageing [1,2,3]. Older people see family relationships, social contacts and activities as valued components of a good quality of life as much as general health and functional status. By having an active social life, older people may fulfil their social needs, which contributes to their wellbeing. By social life, we mean the social contacts people have and the interactions and activities with these social contacts. Both intimate social contacts and more peripheral social contacts contribute to the social wellbeing of older people [3]. If social needs are not fulfilled, this can lead to loneliness and social isolation, which may, in turn, negatively affect a person’s physical and mental health [4,5,6,7,8,9,10]. According to Jong-Gierveld, van Tilburg, and Dykstra [11] loneliness is a subjective emotional state where people experience a lack of (quality) of relationships. The number of existing relationships is smaller than is desirable or admissible, and/or the intimacy one wishes for is not realized. Social isolation is a more objective state. Persons with a very small number of meaningful ties are socially isolated, but they do not necessarily feel lonely [11]. With a growing population of older people in Western society and their need to stay healthy and community dwelling, a healthy and happy social life is especially important [2,12,13,14].

Technology can support the social life of older people, as it does for other areas, and thus contribute to the wellbeing of older people. For instance, domotics facilitate the lives and living circumstances of older people. Domotics range from a simple alarm button in a house to an intelligent system that detects when the client’s activities deviate significantly from their normal daily routines [15]. This is technology that stimulates older people to live independently for longer and to age in place. Most scientific studies about older people in relation to technology primarily focused on the use, attitude and acceptance by older people of technology in general, such as domotics [15,16,17,18].

Technology can also be more social in the sense that it facilitates social contact between people. Email and the Internet can provide various ways for older people to communicate with family and friends as well as provide access to information [15]. This type of technology is called social technology. In this study, we use the following definition of social technology: “any technology that facilitates social interactions and influences social processes between people.” Social technology facilitates social processes through social software and social hardware. Examples of social software are Facebook, email, wikis, blogs, and social networks. Examples of social hardware are devices such as smartphones, tablets and computers but also the landline telephone. 

In this research, we focused specifically on social needs and not on loneliness or social isolation. Focusing on the fulfilment of social needs is a positive and preventive perspective on the social life of older people, instead of only focusing on loneliness or social isolation and on lonely and isolated older people. It has a broader scope and offers a new and fresh perspective for the development and implementation of successful interventions to prevent loneliness and isolation. Studies which focus on the relationship between social technology and the fulfillment of social needs are scarce. Most scientific studies focused on the relationship between social technology and loneliness and/or social isolation. Studies that shed light on the relationship between social technology and its effects on the social life and wellbeing of older people come to diverse, sometimes even contradictory, conclusions. For instance, in the study by Aarts, Peek and Wouters [19], the researchers concluded that a simple association between the use of social technology, in this case, social network sites, and loneliness and mental health could not be determined. In the systematic literature study of Khosravi, Rezvani and Wiewiora [20], 34 technological interventions to reduce social isolation were analysed. They found eight different technologies that have been applied to alleviate social isolation, namely, general ICT (Internet and email), video games, robotics, a personal reminder information and social management system (PRISM), an asynchronous peer support chat room, social network sites, Telecare and a 3D virtual environment. In these eight categories, both effective and non-effective technological interventions were found. Video games and PRISM were the most effective, but both categories contain only one study and it is not reasonable to generalise the effectiveness of those technologies. The findings of the study demonstrate that in principle, each of these technologies can be used to reduce social isolation among seniors. However, more studies are needed to evaluate the effectiveness of new technologies. In the critical systematic literature review of Cohen-Mansfield and Perach [21], the effectiveness of different interventions, both non-technological and technological, for alleviating loneliness among older persons were analysed. Although the interventions analysed are quite diverse, they could be divided into two main groups: one-to-one interventions and group interventions. The researchers concluded that technology in interventions can be effective in reducing loneliness. For instance, technological educational programmes that provide computer training and facilitate the use of a videoconference programme to enable interaction with a family member have proven effective [20]. Sum et al. [22] found positive relations in a population of older people between using the Internet and reduced social loneliness when it was used with friends and family. However, using the Internet for creating new network members resulted in more (emotional) loneliness. The researchers argued that the influence on wellbeing greatly depends on the person you communicate with through the Internet; communicating with unknown people can create more loneliness and anxiety [22]. In the study by Wilson [23], the researcher found that social technology (mostly email) encouraged older people to communicate with friends and relatives. This communication had a positive impact on older adults’ perceptions of self-worth. However, they also found a negative relationship between emotional attachment towards a device (e.g., a smartphone) and a sense of belonging (whether a person feels they belong to society and their network of family and friends). The more the participant was emotionally attached to their technology, the less socially involved they were in their surroundings. This suggests that older people can also be too attached and dependent on their devices, resulting in a negative effect on their wellbeing. 

These findings suggest that although using social technology to improve the social life of older people seems promising, the relationship between technology and the quality of social life can be complex and multi-faceted. Some technological interventions are successful, and some are not, it does not become clear why and which role social technology plays in these interventions. With our focus on social needs and social technology, we offer a broader scope and a new and fresh perspective for the development and implementation of successful interventions to prevent loneliness and isolation. To develop successful technological interventions, further research into the role of social technology as a means to fulfil older people’s social needs and improve their social life is needed. 

### Purpose of the Study and Research Question

This study gives deeper insight into the role social technology can have in the social life of older people. With this study, we provide insight into the current role of social technology in fulfilling the social needs of older people and contribute to the scientific knowledge in this area. This knowledge can contribute to the development and implementation of technological interventions which intend to have positive effects on the social lives of older people. We examined what kind of social technology older people use and why and with whom they use the social technology. We also investigated how social technology contributes to the social lives of the older people who use it, focusing on the barriers and motivations in using social technology. The following research question is central: 


*“What is the role of social technology in the social lives of older people who use social technology?”*


## 2. Materials and Methods

Following approval by the Tilburg University Ethics Review Board (ERB) EC-2018.88, data were collected via 15 semi-structured interviews with older people.

### 2.1. Participants

Data were collected among older people who regularly (more than once a week) use one or more forms of social technology, such as WhatsApp, Facebook, social games such as Wordfeud, Skype, Instagram and email. Participants were selected with the help of the project coordinator. The project coordinator is responsible for the activities and facilities organised in a residence where older people live independently with no or a minimum of care. The project coordinator was asked to select people who, according to their knowledge, use social technology (besides the landline telephone). By the researchers’ use of the snowball effect and visiting a weekly social gathering, older people were invited to participate. Not all older people who were invited wanted to participate; some had health problems or were too busy. Nineteen people were contacted by the researchers, of which fifteen people agreed to participate. Based on previous studies with a similar focus the researchers assumed that 15 participants would be a representative sample [24,25]. Further inclusion criteria for the participants were the following: aged 70 years and older, having sufficient (Dutch) language skills and adequate cognitive and physical abilities to participate in an interview for about one hour. We interviewed people aged 70 and older because at that age, resources, such as health and mobility, sometimes diminish and social networks therefore become more important [2].

### 2.2. Procedure and Materials

After the project coordinator briefly informed the older people of the study and asked if they wanted to participate, the researchers received their names. The selected older people were contacted by phone by the researchers. In this phone conversation, the researchers informed the older people about the study and invited them to participate. When the older people agreed to take part, they received a letter with information about the study. In this letter and in the phone conversation, the older people were told that they could refuse to participate at any time without giving a reason. The participants were asked to sign a letter of informed consent before the interview took place. All the participants received a sensitising workbook at least one week prior to the interview. The sensitising workbook served as a primer to be used by the participants before the interview took place. In the information letter and the phone conversation with the participants, it was mentioned that this workbook helped to prepare them for the interview, but was not obligatory. The researchers tried to make relevant but also fun assignments. The participants were asked the following: (1) to make an ecogram (a drawing) of their social network, (2) to indicate, by means of icons, what form(s) of social technology they use, (3) to indicate reasons for using social technology, (4) to make a schedule of their day and indicate at what time and why they use social technology and (5) to write down tips or ideas for developers and designers to improve forms of social technology. The workbooks were not analysed, but merely served as a primer for the participants so that the older people were well prepared for the interviews since they had already reflected on the role of social technology in their lives.

To answer the research question “What is the role of social technology in the social lives of older people who use social technology?”, we conducted semi-structured interviews during which we used a topic list addressing the following topics: Attitude (how do you feel about social technology?). We ask why and how they start using social technology (start using technology). We asked what, with whom (network), but especially why (reasons) older people use social technology. We especially asked about the situations in which older people use social technology. The topic list for the semi-structured interviews is based on the results of two systematic literature studies, one on the social needs of older people and one on the factors influencing the acceptance of technology [17,26], and two qualitative studies on the social needs of older people and the role of social technology [24,25].

Two interviewers (a researcher in the area of applied psychology and a student in his final year of the Bachelor of Applied Psychology programme) conducted the interviews.

After ten interviews, hardly any relevant new information related to the purpose and research question was extracted. After fifteen interviews, data saturation was achieved.

### 2.3. Analysis

With the permission of each participant, interviews were digitally audio-recorded and transcribed verbatim. Based on the transcripts, the researchers made a personal profile for each participant and a list of all the reasons why the participant used social technology. A thematic analysis [27] was applied using a combination of inductive and deductive coding [28,29], using qualitative data-analysis software (Atlas.ti version 8).

In the first phase of the (open) coding process, two researchers were involved in the coding process, to ensure objectivity [27]. The researchers attached the inductive codes first individually and at a later stage together and discussed the discrepancies to reach a consensus. The agreement between the researchers was high; the discrepancies in the coding process were small and only related to different names for the same fragment/concept.

In the second phase (axial coding), we used the following sensitizing concepts based on previous studies [23,24,25] to interpret the data. These concepts are (1) first contact with social technology, (2) contact with network members (intimate and peripheral), (3) reasons for using social technology, (4) advantages and disadvantages of social technology, (5) what are they using (hardware and software), (6) characteristics of the respondent, including attitude, (7) barriers in use.

In the third and last phase (selective coding), the three researchers discussed which themes best represent the data and reflect the research question. The researchers especially looked at ‘What is the role of social technology in the social lives of older people who use social technology?’. Five themes emerged from this analysis, three describing the role that social technology plays in the social lives of the participants and two describing the downside of using social technology: (1) strengthening social relationships, (2) enriching social contacts, (3) reassuring older people and their network, (4) usability and (5) ambivalent attitude towards social technology.

## 3. Results

In Table 1 the characteristics of the participants are presented. In total, fifteen people, of whom nine are female, participated in our study. The age range is from 70 to 92 years. The sensitising workbook was completed by six of the participants. Some other participants only read the assignments to be inspired by them but did not actually carry them out. The rest of the participants did not do the assignments in the workbook. The social technological devices (hardware) used most often by the participants are the landline telephone, the smartphone and the tablet. Most of the participants frequently use their landline telephone to contact network members. They prefer this telephone sometimes because it is bigger and easier to use than a smartphone and they are accustomed to using it. The social technological applications (software) most used by the participants are WhatsApp and email. Most of the participants were made enthusiastic about social technology by children and grandchildren; our participants realise that to communicate with and contact them, social technology can be a resource. Some of our participants are interested in and using technology because they had a technical profession.

The role that social technology plays in the lives of the participants and the time they spend using social technology varies. Most of the participants use social technology every day because it enables them to communicate more easily with their friends and family. Some of the participants even pointed out that they could not live without social technology. 

The five themes that emerge from our analysis are (1) strengthening social relationships, (2) enriching social contacts, (3) reassuring older people and their network, (4) usability, and (5) ambivalent attitude towards social technology.

### 3.1. Strengthening Social Relationships

For our participants, social technology strengthens the social relationships they have with both their intimate and their peripheral network members. Social technology makes communication with existing network members easier and therefore, more frequent. One participant said, 


*“My sister lives far away, and we don’t visit every day or every week and then WhatsApp is just really precious”*
(Woman, 72)

Most of the participants use social technology to strengthen existing social structures and social relationships. Only one of the participants uses social technology to make new contacts; she plays an online game (Wordfeud) with people she does not know and chats with them. However, the relationship with these contacts remains superficial and only relates to playing Wordfeud. Some of the participants also indicated that social technology facilitates the arrangement of face-to-face contact. For example, one of the participants said, 


*“I meet my sister every week to chat and see how she is doing […], we send each other a message in the WhatsApp saying: Hi sister, where shall we meet this time?”*
(Woman, 83) 

Many participants have friends and family who live, study and work all around the country and even abroad; social technology facilitates and strengthens contact with them. One of the participants commented that social technology, especially WhatsApp, strengthens the connectedness and bond in his family. A few times during the interview, he pointed out that social technology is an important communication and bonding tool in his family. It makes the communication in his family more easy, frequent and fun. He said,


*“If we didn’t have this (WhatsApp), we could not have built our bond of trust in the family.”*
(Man, 92)

One of the older participants has a brother who lives in Canada and is going through a difficult time. She offers him emotional support using social technology. She said,


*“I speak to my brother almost every day, mostly through skype […], My brother is single and at the moment he is not doing well. He needs his big sister, he’s four years younger than me.”*
(Woman, 82)

Social technology can be instrumental in the sense that it enables older people to communicate more easily. It also can be more emotional in the sense that through social technology, help and comfort can be offered to loved ones. Social technology offers possibilities in providing social support to network members when face-to-face contact is not possible. However, most of the participants prefer face-to-face contact to strengthen their relationships and only use social technology when face-to-face contact is not possible. One participant said,


*“It’s all about the real contact, the warmth in our family”*
(Man, 92)

One of the participants uses the technology to gain insight into the activities and wellbeing of his family. He has files on his computer with information and correspondence of all his children and grandchildren. It gives him a sense of peace and even a sense of control that he knows what everybody is up to. He said,


*“I like to know what everyone is up […] We have an agreement in the family that everyone responds to each other within 5 min through WhatsApp”*
(Man, 92)

He enjoys looking at the information about his family; it brings back happy memories. He showed the researcher a digital album of the 65th wedding anniversary of him and his wife and said,


*“This is my family, these photos are all in the cloud, and can always be reached there, everything is about family, that is wat is most important”*
(Man, 92)

The fact that he has this information and precious memories on his computer strengthens the already close family ties.

Social technology occasionally facilitates communication with organisations for volunteer work. One of the participants uses social technology (i.e., email) to receive information about the clients she has to visit for her volunteer work. The use of social technology can also strengthen feelings of being useful. This is illustrated by the following quotes:


*“I like that because of my smartphone people can contact me at any time when they are in need of help”*
(Woman, 82)


*“Because of my smartphone they all know where to find me”*
(Man, 70)

This participant often helps his neighbours and friends with chores and is, because of his smartphone, easy to contact. Therefore, social technology strengthens the relationships our participants have with others and makes communication easier, for example, with possible volunteer work. It also strengthens contact with friends and family by enabling them to offer emotional support. Social technology in this matter connects the participants to other people, to the community and to society and fulfils the need to feel meaningful.

### 3.2. Enriching Social Contacts

For our participants, social technology not only strengthens their social relationships but also enriches the moments of social contacts. All participants connect with their social relationships through sharing verbal (stories) and visual (photos/videos) information with their families and friends. Stories in the form of updates of one’s life vary from light and funny stories to the sharing of deeper emotional feelings and experiences. For example, one participant said,


*“A friend of mine, she is 80 years old and she recently send me a really funny video about people in their seventies who cannot go outside because of the cold […] and they have to make up things to do inside [..], it was really funny”*
(Woman, 82)

The participants make jokes and send short messages through WhatsApp, but, through Skype, they have deep emotional conversations, as the participant has with her brother living in Canada. In addition to sharing verbal information, almost all participants enjoy sending and receiving photos and videos, for which they mostly use email and WhatsApp. The photos that older people take and share within their network are often related to the personal experiences of the participants. For instance, one of the participants loves to go for walks in nature and sends pictures to her children and friends of the beautiful landscapes she encounters. The morning of the interview, it was sunny and foggy; she said,


*“It was so beautiful and mysterious, that fog, I had to make a picture and send it to my friend.”*
(Woman, 77)

A good friend of one of the participants, an older woman, moved to Turkey, where they are building a house.


*“The man is an artist […], look at this photo, he did everything himself, the paintings on the wall are like icons, and when I watch this video he send I can see him work”*
(Woman, 78)

The older woman regularly receives pictures and videos of the house. In this way, she is able to keep in touch with her friends and stay informed of their situation. In turn, this woman sends her friends pictures from her own life. The participants also can see the places where their network members go on vacation more easily and be more connected and more aware of the lives of these network members. They become a part of the holiday experience. One participant said,


*“I like the fact that when they (my children) are on a holiday, they immediately send pictures and I don’t have to wait to see the pictures until they return.”*
(Woman, 82)

Social technology enriches social contacts and makes older people more involved in the lives of their loved ones and vice versa. It gives the participants joy to share visual materials of their lives and activities and to have visual images of what family and friends are doing, especially when these friends or relatives are further away, making other forms of communication difficult.

Occasionally, there are practical reasons for sharing photos. One of the participants said,


*“My granddaughter now has an important job […], I said you need a car, let me help you and then she was in the showroom and sent me a photo […], I said that is the one you should take”*
(Man, 92)

Another participant said,


*“I love making cryptograms, on Saturday there is a large cryptogram, my friend who lives in […] makes it at the same time, then we send each other WhatsApp messages: 9 letters? Do you know?”*
(Woman, 83)

Through WhatsApp, they communicate about the puzzle, help each other and make jokes. In this way, the use of social technology thus enriches her social life.

Social technology also serves as an important storage point for social interactions and memories. Older people enjoy looking back at photos and conversations they had with their friends and family.

A clear advantage of social technology (except for the landline telephone) in comparison to face-to-face contact is that the social interactions are saved and therefore, can be watched and read at any time. One participant carefully stores all the photos she takes and receives from her family in a digital album; she said,


*“All those photos, those will never get lost, they always stay there”*
(Woman, 72)

Therefore, social technology enriches the social lives of our participants because it enables them to feel connected to their loved ones even when there is no direct contact or communication.

### 3.3. Reassuring Older People and Their Network

Social technology and the fact that through their smartphones, the participants are always connected, offers them a sense of security and peace of mind. They can communicate at any time with their social relationships, which is very reassuring for the older people as well as for their family and friends. The children of some of our participants encouraged their parents to purchase a smartphone for their own peace of mind as it is an easier way to be reassured that their parents are in good health. One participant said,


*“They can always reach me now, which is of course really important. For my daughter, it is a relief that I can always call her.”*
(Man, 92)

This was also the reason why one of the participants purchased a smartphone. When he had car trouble years ago, he decided to buy a smartphone. Now, he can contact network members at any time and that gives him a feeling of reassurance. Most of the participants mentioned that when having the smartphone with them, they feel safer and at peace.

The participants in our study also use social technology to facilitate and structure their daily lives. It helps them to remain more independent and in control of their own social life. One of the participants uses her smartphone as a calendar and reminder for appointments with her network members. She said,


*“I use the tablet and its schedule to be in touch with the people I know. I know when I last saw them, if it was face-to-face or through telephone and when our next meeting is.”*
(Woman, 72)

Social technology does not only give a sense of control, safety and structure, it also makes it easier for participants to ask for help if it is needed. Social technology facilitates communication to help network members and reassures both the older people and their social contacts.

### 3.4. Usability

Although social technology strengthens and enriches the social lives of our participants and provides reassurance, they sometimes feel frustrated and experience barriers in using social technology. They struggle with the usability of social technology. For instance, all the participants indicated that they sometimes encounter technological problems with their devices and have trouble using it. They struggle with updates, passwords, Wi-Fi, and how to use the devices and the applications. One of the participants said,


*“We are not always friends, me and my smartphone.”*
(Woman, 83)

She further commented:


*“[…] and then I am struggling with it and there is nobody at home to help me, that is really frustrating, That I phone really can be a pain in the ass, and then I think, I will throw the thing out of the window”*
(Woman, 83)

Some of the participants only passively use the photo options on their devices; they receive photos from friends and family, but find it difficult to take, send or forward photos themselves. They do not understand how this option works, or they forget how to use this function. One of the participants only recently started using a tablet that she received from her son. She was clearly struggling with how to use and benefit from it. She said,


*“I pushed all the buttons, because I didn’t know what to do.”*
(Woman, 92)

This participant is willing and eager to use the tablet, but without help, she simply cannot. When the participants face difficulties in using the technology, they often ask their children or grandchildren to help them. In these cases, social technology is a reason for contact with children and grandchildren. Another participant said,


*“When I don’t understand how to use the smartphone, I ask one of my two sons to help out.”*
(Man, 92)

Overall, the participants ask for help relatively easily. Most of them frequently ask a family member or friend for help when they struggle with social technology. Sometimes, those friends or relatives live further away, which makes asking for help more difficult.

Almost all the participants indicated that they need help from family or friends to be able to use the social technology. Therefore, network members are needed to be able to use the social technology to get in touch with (other) network members. In other words, the social network of the participants is a condition for using social technology.

### 3.5. Ambivalent Attitude towards Social Technology

In addition to barriers in use, the participants also expressed an ambivalent attitude towards social technology. Some participants expressed that social technology is not as personal as other forms of communication. One participant said,


*“These days one can send cards digitally, but I much more prefer handwritten postcards, that’s so much nicer”*
(Woman, 78)

According to this participant, it takes more effort to send a postcard. This participant uses social technology and clearly sees the benefits, such as the ease of communication and the feeling of safety, but she states that if possible, she prefers face-to-face contact and real postcards. She has a clear view of how and when she uses social technology and when she prefers other ways of communication. This attitude is shared by other participants; they use social technology when other ways of communication are not possible. Some participants dislike the fact that social technology sometimes stands in the way of or replaces real human contact, for example, when sitting at a diner and everyone is looking at their smartphone instead at looking and talking to each other. They see network members and other people spending too much time on their smartphones. One of the participants said,


*“All day, those sounds: ping ping. When my children visit, I say, ‘Please turn off your phone when you are at my place, I don’t like that.’”*
(Woman, 78)

This participant makes comments to her (grand) children that when they are at her place, they should have real face-to-face conversations and not stare at their smartphones. When the grandchildren visit their grandmother, they know that she wants their sincere and full attention.

Another participant is afraid that she will be excluded because of the modern technologies. She is really motivated to keep up and sees the possibilities of technological developments but sometimes feels that it is all moving too fast for her generation. She points out that she prefers real face-to-face contact but that for younger people, this seems less obvious. She said,


*“And it (the modernisation of society) is still continuing and if you don’t keep up with that you will be excluded.”*
(Woman, 74)

Overall, the participants feel a need to keep up with their families and friends and see that as an important motivation to use social technology. One participant also pointed out the necessity of using WhatsApp to contact her grandchildren.


*“When you don’t use WhatsApp, you never hear from them.”*
(Woman, 78)

The participants sometimes feel pressure to keep up with their relatives and with society and although they do not always agree with the amount of time their network members spend on their devices, in general, they are motivated to use social technology.

## 4. Discussion

In this qualitative research, the role of social technology in the social lives of older people was studied. The role of social technology in the lives of older people is that it strengthens existing social relationships, it enriches social contacts and it reassures both the older people and their network members. Social technology, in this matter, can be seen as a friend of older people; it makes their social lives stronger, richer and more at peace. Social technology is a frenemy in the sense that besides all the advantages, older people have problems in using social technology and have an ambivalent attitude towards social technology.

The fulfilment of social needs contributes to a healthy and happy social life and to the social wellbeing of older people. Looking at the three social needs identified in the study of Bruggencate et al. [25]—connectedness, meaningfulness and independence—social technology in our study helps to fulfil all three needs, but especially the need for connectedness by strengthening and enriching social relationships. To strengthen and connect, older people share stories, memories, emotions and photos by means of social technology. These interactions vary from funny and light to deep and emotional. Sharing visual information (photos and videos) with friends and family is very popular among the older participants in our study; these benefits of social technology are appreciated and differ from communication in the past where only verbal communication was possible through the landline telephone. The possible role of social technology in creating (social) connectedness is also highlighted in the study by Sinclair and Grieve [30]. In this study, Facebook created social connectedness in a population of older people. In the study of Barbosa Neves, Franz [31] social technology especially increased social connectedness with geographically distant relatives. In the current era of globalization, social technology is a solution to strengthen social ties when loved ones are living and working further away.

Social technology reassures and gives peace of mind both to older people themselves and their friends and relatives, but also gives a sense of structure and sometimes control. The participants feel independent and autonomous when using social technology and when connecting with network members and society. The relationship between social technology and independence is, however, somewhat ambiguous. On the one hand, it helps to facilitate and have control over one’s life, while on the other hand, it offers older people a sense of dependence because they have problems using it and forces them to depend on others to solve those issues. All the participants in our study experienced barriers in using social technology and depend largely on others for help. That there can be an ambivalent relationship between technology and wellbeing also becomes clear in the study by Wilson [23], where while positive relations between using technological devices and self-worth were found, older people can also depend too much on the technology, which causes a reduced sense of belonging.

The participants in our study indicated that by using social technology, it is easier to engage in and communicate with their volunteer work. Social technology also enables them to offer emotional support to their friends and family. Social technology connects the participants to a community and to society and in this way, fulfils the need to feel meaningful. The researchers of one study [25], concluded that (social) technology can indeed enable older people to engage more easily in volunteer work; it can also be used to share stories and experiences and to offer support and comfort. As these researchers also concluded, by staying active in a meaningful way, all three needs can be fulfilled, the need for connectedness, independence and meaningfulness [26].

Family plays an important role in the acceptance and use of social technology, as also indicated by Luijkx et al. [24] and Peek et al. [16]. The older people in our study found it relatively easy to ask for help and often have a relative who helps them, but this is not always the case. This is in line with the study by Peek et al. [16], which illustrated that support and coaching may be essential in the adaptation and use of technology by older people. Barriers older people face in using social technology are mostly congruent with those described in the systematic literature study of Peek et al. [17]. In their study, 27 factors divided into six themes were identified that influence the acceptance of technology in the pre-implementation stage; one of these themes is ‘concerns about technology’. These concerns are high costs, usability and privacy implications. All three concerns are mentioned in our study, especially usability. All the participants at one time struggled with using the devices and social technology. As one of the participants said about her smartphone, “We are not always friends”. The struggle of the participants is significant, which is why sometimes, social technology feels more like a frenemy than a friend for the participants. The role of social technology is that it enriches, strengthens, reassures, but it also and to a large extent, frustrates our participants. As we mentioned at the beginning of this paragraph, the participants depend largely on family and friends to help them out with social technology. Studies show that older people are not eager to ask for help or to be dependent on others [32,33]. For the participants in our study, asking for help was not mentioned as a barrier. 

In addition to the struggle of usability the participants face, their attitude towards social technology is mostly positive. The advantages seem to outweigh the disadvantages. There is a shift of attitude towards social technology when older people start to use it, as also described in the study of Bruggencate [24]. In this study, the participants who do use social technology are enthusiastic. The participants who do not use social technology are often negative or do not see any benefits in using it.

Although, in this study, we did not explicitly look for information about what influences older people’s purchase and use of social technology, this topic was addressed by some participants in relation to their attitude towards technology and their reasons for using it. Our findings underline the most important aspects of existing technology acceptance models [34,35,36,37,38] in that perceived usefulness and ease of use are the most important influencing factors. Basically, older people will use social technology if the benefits of using it are clear to them and if it is not too difficult to use (or help is present).

Looking at the role social technology plays in the lives of the participants, most older people could probably benefit from using it. However, it seems to be especially used by people with an already existing network and enough social and technological skills. This also becomes clear in the study by Hage [39]. In her thesis, Hage [39] argued that online communication strategies as interventions to create connectedness are mostly only beneficial for highly educated, rich and younger (than 65 years) older adults. Hage [39] even argued that the implementation of online communication technologies often increases the social inequality between vulnerable older adults and non-vulnerable older adults. Based on our study and other studies [23,26,31], using social technology for creating new friendships is not a solution for their loneliness or isolation, particularly for vulnerable older adults. It is probably better to first create a social structure and network; social technology can then strengthen these relationships.

Older people seem to have a thoughtful way of looking at and using social technology. For them, it should not replace face-to-face contact. Spitzer [40] argued that social skills are better taught through face-to-face contact [41]. Turkle [42] also warned against the excessive use of technology, especially in the social area. According to Turkle [42], there is a great risk of relations becoming more superficial when people communicate through technology. The older people in our study do not use social technology to make new contacts; this seems wise considering the results of Sum et al. [22] that showed that loneliness increases when social technology is used to make new contacts.

### Limitations

We realise that the benefits and barriers these older people face in using social technology are not generalisable to the whole population of older people. The participants of our study are older people who live independently with a minimum of care and who regularly use some form of social technology besides the landline telephone. This means that they are relatively healthy and have the motivation and the financial means to use social technology. Furthermore, the older people in our study all have a network of friends and family; almost all of them receive help in using social technology. For people with a smaller network, this help may not be available.

All the participants received a workbook prior to the interview, which served as a primer. It gave them the opportunity to think about how and why they use social technology. Almost half of the participants did not complete the workbook, because they forgot, were too busy or just did not feel like doing this. The participants who did complete the workbook were better prepared for the interview topics and could discuss these more easily than the others. As a result, the interviews with these participants had more structure and depth. For future research, such primers are highly recommendable. Researchers can stimulate participants to use the primers by creating interesting assignments and preparing well designed workbooks.

## 5. Conclusions

Based on the results of our study, we can conclude that social technology mostly is a friend of older people; it makes their social life stronger, richer and more at peace. Social technology is a mean or recourse to connect the older individual to their network members and to society. Social technology strengthens the existing social relationships and structures of older people and brings both depth and fun to social relationships. The fact that visual and verbal interactions can be saved and re-watched is a great advantage. Social technology offers the participants structure and control on the one hand but makes them feel dependent on the other hand. Most participants struggle with using social technology and need help from their network. They also feel that social technology sometimes stands in the way of real human contact. However, with proper support, social technology can play an important role in the lives of older people, primarily in facilitating and strengthening their existing social relationships. Interventions can therefore best focus on facilitating and supporting older people’s use of social technology with existing network members. We conclude that social technology can indeed be a good friend to a large group of older individuals, a complex friend with a high maintenance, but a friend they would not like to miss.

## Figures and Tables

**Table 1 ijerph-16-04969-t001:** Characteristics of participants and their use of social technology.

*Participant*	*Sex*	*Age*	*Social Technology Hardware*	*Social Technology Software*
1	F	77	Landline telephone, smartphone, tablet, laptop	WhatsApp, Facebook, email
2	M	92	Landline telephone, smartphone, tablet, laptop	WhatsApp, email
3	M	92	Landline telephone, smartphone, tablet, laptop	WhatsApp, games (chess)
4	M	92	Landline telephone, tablet	Email, Skype
5	F	83	Landline telephone, smartphone, tablet	WhatsApp, email
6	M	90	Landline telephone, smartphone, tablet, PC	Email
7	F	78	Landline telephone, smartphone, tablet	WhatsApp, email
8	F	82	Landline telephone, smartphone, tablet, PC	WhatsApp, Facebook, Skype
9	F	92	Landline telephone, tablet	Skype
10	F	72	Landline telephone, smartphone, PC	WhatsApp, Facebook, email
11	F	72	Landline telephone, smartphone, laptop	WhatsApp, Facebook, email, Skype
12	M	74	Landline telephone, smartphone, tablet, PC	WhatsApp, Facetime, email
13	F	78	Landline telephone smartphone, tablet	WhatsApp, Facebook, email
14	F	74	Landline telephone, smartphone, laptop	Facebook, online games
15	M	70	Landline telephone, smartphone, tablet	WhatsApp, Facebook, online games

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
