# Peer review of "Friends or Frenemies? The Role of Social Technology in the Lives of Older People"

_ijerph, 2019, doi:10.3390/ijerph16244969_

Round 1

Reviewer 1 Report

This is an interesting study from the perspective of older adults themselves on their use of technology. I am not convinced by the use of the primer if it was not analysed or used in any other way but to prepare the participants. Were they aware it was not going to be used? As this raises ethical questions for me around collecting information that is not used in the analysis.

 There also needs to be a clear definition of both loneliness and social isolation. The introduction talks a lot about loneliness so it appears that this is a theme that runs through the paper and sets the scene that technology has a role in reducing loneliness and social isolation. Yet, later it is apparent that this is not the case. This needs to be clearer and the focus on loneliness needs to be reduced in the introduction if this is not the authors intention to focus on loneliness

The biggest issue is there are not enough quotes from the participants, and those that are used are not that informative. An example is where one discuss one participant keeps files on his pc, the authors talk about a sense of control. It would be nice to see a quote from the participant which illustrates/supports this point.

 I feel the analysis presented is not deep enough to support the conclusions and this would be helped by additional quotes and reducing text elsewhere to keep within the word limits.

More discussion of the barriers/issue older adults reported would also be helpful as at the moment it appears there is a lack of balance in the reporting with more emphasis on the positive and less on the negative.

Reviewer 2 Report

Summary

The current study seeks to better understand the role social technology plays in shaping the lives of older adults. Because most research on social media and other technologies has used younger adult samples, the current study helps provide insight into how communication technologies affect the lives of older adults. I enjoyed reading the manuscript and believe it is well-written. However, as detailed in my comments below, important information about the methods and data analysis used in the current study is missing from the manuscript, making it difficult to judge the reliability and validity of the qualitative data.

Comments

P. 3 – The sample size is only 15. How was that number decided? For the qualitative data, is there any way to ensure that was enough to achieve data saturation? Would it be helpful to have older adult non-users of social technology in your sample as well? That might help better isolate their attitudes towards social technologies to contrast with the users’ attitudes? P. 4 – More description of the semi-structured questions is needed. Without knowing what the specific prompts were, it is hard to judge the themes that are reported in the Results section. P. 4 – Inter-rater reliability is mentioned, but no metric or result is given. Was it percent agreement or some other metric? How much discrepancy was there between the two coders that needed to be resolved? P. 4 – It just says that thematic analysis was used. But how were the codes generated? Were the codes/themes based on top-down or bottom-up coding? A mixture of both? What were the frequencies of the themes? For example, how many of the 15 participants provided responses that were coded as the five themes that are reported? How was the determination of a theme made? Were other themes identified but didn’t reach some specified cut-off level of frequency? Without knowing these rules, it is very difficult to interpret the reported themes in the Results section. Were the identified themes consistent or inconsistent with any technology acceptance models? That is, it would be helpful to know more about any theoretical contributions the current results have on our understanding of technology acceptance and usage by older adults.

Round 2

Reviewer 1 Report

The authors have responded to the comments and substantially improved the paper. I am happy for this to be accepted in its current form.

Reviewer 2 Report

I believe that the authors did a nice job addressing the issues raised by the reviewers. I still think the soundness of the qualitative data analysis would be improved if a numerical account of inter-rater reliability was provided, as well as some type of frequency count that makes it clear how many of the 15 participants expressed each of the themes/codes. But I do believe that the manuscript has been greatly improved and is more suitable for publication now. Nice work!